# Impact of Activity Monitoring on Physical Activity, Sedentary Behavior, and Body Weight during the COVID-19 Pandemic

**DOI:** 10.3390/ijerph18147518

**Published:** 2021-07-15

**Authors:** Jacob E. Barkley, Gregory Farnell, Brianna Boyko, Brooke Turner, Ryan Wiet

**Affiliations:** 1School of Health Sciences, Kent State University, Kent, OH 44242, USA; rwiet@kent.edu; 2Department of Exercise Science and Sports Studies, John Carroll University, University Heights, OH 44118, USA; gfarnell@jcu.edu (G.F.); bboyko21@jcu.edu (B.B.); bturner@jcu.edu (B.T.)

**Keywords:** exercise, sitting, coronavirus, fitness tracking

## Abstract

Decreases in individuals’ physical activity and increases in sedentary behavior and bodyweight have been reported during the COVID-19 pandemic. The present study assessed the ability of physical activity monitoring, which may promote physical activity and discourage sedentary behavior, to mitigate these negative outcomes. An evaluation of university samples (*N* = 404, 40.5 ± 15.4 years) of self-reported physical activity, sedentary behavior, and bodyweight prior to the closure of campus due to the pandemic in March of 2020 and again at the time of the survey administration (May–June 2020) during pandemic-related restrictions was performed. Participants also reported whether they did (*n* = 172) or did not (*n* = 232) regularly use physical activity monitoring technology. While physical activity was unchanged during the pandemic (*p* ≥ 0.15), participants significantly increased sitting by 67.8 ± 156.6 min/day and gained 0.64 ± 3.5 kg from pre-campus to post-campus closure (*p* < 0.001). However, the use of activity monitoring did not moderate these changes. In conclusion, while physical activity was not affected, participants reported significant increases in sedentary behavior and bodyweight during the COVID-19 pandemic. These changes occurred regardless of whether participants regularly used physical activity monitoring or not.

## 1. Introduction

The coronavirus disease 2019 (COVID-19) is a respiratory illness that is transmitted from person to person via respiratory droplets and by 2020 it induced a global pandemic [1,2]. In an effort to curb the spread of COVID-19 there were guidelines established by the spring of 2020 that limited access to fitness facilities, outdoor recreation areas, and other businesses and public spaces. These changes likely limited access to physical activity options for many individuals which may result in reductions in physical activity behavior [3,4]. Additionally, recommendations to socially distance from others outside of one’s home likely altered social interactions. This may also negatively impact physical activity behavior since positive social interaction is predictive of physical activity participation [5,6,7,8,9]. Early in the pandemic, several researchers outlined how this environment created by the pandemic may have negatively impacted physical activity and sedentary behavior [10,11,12,13,14,15]. These researchers also provided recommendations for methods to maintain physical activity (e.g., at home exercise) in this challenging environment in an effort to avoid these potentially negative outcomes.

Despite these warnings and recommendations, emerging research has indicated that the pandemic-related restrictions may have negatively affected physical activity and sedentary behavior [16,17,18,19,20,21,22,23]. Much of this research has indicated that individuals are sitting >15% more during the pandemic than they were before it began [16,20,21]. While somewhat equivocal, there is also mounting evidence that some individuals have also decreased physical activity behavior [20,21,22]. For example, Maugeri et al. (2020) reported >30% reductions in physical activity during the pandemic in sample of adults in Italy [22]. However, research from our group and others has reported that those who were most active before the pandemic reported the greatest reductions (20–30% decrease) in physical activity during the pandemic whereas individuals who were less active pre-pandemic had smaller reductions (2% decrease) or, in some instances, increases in physical activity (>14% increase) [20,21].

While there is some equivocation regarding changes in physical activity, if the pandemic has created an environment which promotes greater sedentary behavior and less physical activity, it is possible that weight gain would occur [24]. Recent research supports this notion and has provided initial evidence of weight gain during the pandemic [23,25]. For example, Bhutani et al. (2021) reported average weight gains of 0.64 kg during the pandemic [25]. These researchers also reported that 40% of those surveyed reported gaining weight and that >50% of that group reported gaining >2.3 kg. While reductions in physical activity, increases in sedentary behavior, and weight gain are all associated with increased risk of a variety of cardiometabolic diseases, such changes may be particularly concerning as it pertains to the current environment [26,27]. Elevated bodyweight and reduced fitness have emerged as risk factors for becoming seriously ill should an individual contract COVID-19 [13,28,29]. Therefore, further examinations of the impact of the pandemic upon these negative behavioral changes and methods to potentially mitigate them are of importance. Not only will this information be of use in places where the COVID-19 pandemic persists, but it could also aid in developing improved responses for similar events in the future [30].

One possible method to prevent decreases in physical activity and increases in sedentary behavior is the use of physical activity monitoring technology. The use of such technology has been associated with greater physical activity and decreased sedentary behavior [31,32,33]. Work from our group also suggested that using activity monitoring technology aids in the maintenance of physical activity behavior in women [34]. However, there is conflicting evidence that suggests the use of activity monitoring may simply be a marker for an elevated exercise identity and that this technology does not provide any benefit as part of an intervention designed to increase physical activity and decrease sedentary behavior [35,36]. Therefore, evidence examining the benefits of using physical activity monitoring technology is equivocal.

The purpose of the present study was two-fold: (1) To assess changes in physical activity, sedentary behavior, and bodyweight from pre-COVD-19 to during the COVID-19 pandemic and (2) to assess the ability of physical activity monitoring technology to moderate any pandemic-related changes in physical activity, sedentary behavior, and bodyweight. The present study hypothesized that there will be significant decreases in physical activity and increases in sedentary behavior and bodyweight from pre-pandemic to during the pandemic [16,17,18,19,20,21,22]. Additionally, while there is some evidence supporting the utility of physical activity monitoring for promoting physical activity, others have indicated that such technology does not aid in this regard [31,32,33,35,36]. Therefore, we hypothesize that the use of physical activity monitoring technology will not moderate the effects of the pandemic. In other words, the use of physical activity monitoring will not prevent the hypothesized negative behavioral changes during the pandemic.

## 2. Materials and Methods

### 2.1. Participants

This study consisted of a sample of a population (i.e., faculty, staff, undergraduate, and graduate students) from Kent State University (Kent, OH, USA). A link to an anonymous online survey was emailed to all university faculty and a random selection of university students (both graduate and undergraduate) from the study’s PI (Barkley). A link to the survey was also provided in an email newsletter which was sent to all university staff. The survey was available from 18 May 2020 through 3 June 2020. A total of *N* = 714 surveys were returned, making up the initial sample. Participants that did not complete the entire survey for the variables of interest (e.g., physical activity, bodyweight, etc.) were eliminated from the study, resulting in a new sample of *N* = 404 participants. Participants were asked to report their age, gender, and university role. That final sample had an average age of 40.5 ± 15.4 years old and consisted of *n* = 291 females, *n* = 110 males, and *n* = 3 non-binary/choose to self-describe. There were *n* = 97 undergraduate students, *n* = 81 graduate students, *n* = 175 faculty, *n* = 29 staff, and *n* = 22 administrators/other.

A priori power analyses were performed using the results of previous survey-based research examining changes in sedentary behavior and physical activity pre-pandemic relative to during the enaction of pandemic restrictions [16]. This previous study reported a 58% increase in sitting and a 38% decrease in physical activity during the pandemic. These changes yielded effect sizes of Cohen’s *d* = 1.13 and 0.28 for sitting and physical activity, respectively. Given these effect sizes and an α ≤ 0.05, samples of *n* = 10 and *n* = 98 participants would be needed to achieve a power ≥ 0.80 for differences in sedentary behavior and physical activity, respectively. Given these analyses, the final sample of *N* = 404 was adequate.

### 2.2. Procedures

Within the anonymous online survey, after reporting their age, university role, and gender, participants then self-reported their physical activity, sedentary behavior, and bodyweight before the cancellation of face-to-face classes and again at the time the survey was completed. All face-to-face classes were cancelled across the university on 11 March 2020. On 20 March 2020, all university facilities were closed and students were sent home. Shortly thereafter (22 March 2020) the Ohio governor issued stay-at-home orders which persisted until 18 May 2020. At that time, the guidelines were altered to a “strong recommendation” for social distancing and a continued ban on gatherings of >10 people [37]. These closures and restrictions likely significantly altered the daily lives of those participants in our sample and the purpose of this study was to assess those effects. Assessing past and present physical activity and sedentary behaviors using survey methods is valid and has been implemented previously [38,39,40,41,42,43,44]. All participants provided informed consent prior to completing the survey and all study procedures were approved via the university institutional review board.

Physical activity was assessed via the validated Godin physical activity questionnaire [45,46]. This survey assesses the number of times per week that individuals report participating in 15 min of strenuous (i.e., vigorous), moderate, and light physical activity. The score for each intensity is calculated using the following equations: times per week participating in strenuous ×9, moderate ×5, and mild ×3. The values for each intensity are summed for a total physical activity score. This scale possessed good internal consistency (Cronbach’s α = 0.77) in the present study.

Participants were asked to report their typical physical activity at two times: prior to the cancellation of face-to-face classes (11 March 2020) and again at the time the survey was completed. For assessing physical activity prior to cancellation, the following language was utilized: “During a typical 7-day (one week) period before Kent State ended face-to-face classes (11 March 2020) how many times on the average did you do the following kinds of exercise for more than 15 min during your free time?” For reporting their physical activity at the time they completed the survey, the following language was utilized: “During a typical 7-day (one week) period after Kent State ended face-to-face classes (11 March 2020) how many times on the average do you do the following kinds of exercise for more than 15 min during your free time? In other words, these questions are asking you to describe your current physical activity.”

Sitting time was assessed using the validated International Physical Activity Questionnaire (IPAQ) [39]. Participants were asked “During a typical week before Kent State ended face-to-face classes (11 March 2020), how much time did you usually spend sitting on a weekday?” The same question was then asked for sitting during the weekend. Participants were then asked to report their current sedentary behavior using the following language: “During a typical week after Kent State ended face-to-face classes how much time do you usually spend sitting on a weekday/weekend?”

Participants were asked to report their bodyweight before the cancellation of face-to-face classes and their current bodyweight. There is evidence supporting the validity of self-reporting bodyweight in adults [47,48].

Lastly participants were asked to respond “yes” or “no” to the following question: “Do you regularly use a physical activity monitor (for example, Fitbit, Nike Fuel Band, activity tracker apps, pedometer, etc.)?” From this question, two separate groups were established: Those who used physical activity monitoring technology (*n* = 172) and those that did not (*n* = 232).

### 2.3. Statistical Analyses

Multiple two group (used monitoring technology and no monitoring) by two time points (pre pandemic and post cancellation of face-to-face classes) analyses of variance (ANOVAs) with repeated measures on time point were used to assess changes in mild, moderate, vigorous, and total physical activity; sedentary behavior; and body weight. When necessary, post hoc analyses of any significant effects were performed using *t*-tests. A priori significance was set at α ≤ 0.05 and all data were analyzed using SPSS Version 26 (IBM Corp., Armonk, NY, USA).

## 3. Results

### 3.1. Total Physical Activity

There were no significant main (F(1, 402) ≤ 0.86, *p* ≥ 0.36, partial η^2^ ≤ 0.001) or interaction (F(1, 402) = 2.6, *p* = 0.11, partial η^2^ = 0.006) effects for differences in the total physical activity (Table 1).

### 3.2. Vigorous Physical Activity

There was a significant main effect with respect to group (F(1, 402) = 10.2, *p* = 0.002, partial η^2^ = 0.025) for differences in vigorous physical activity. Those participants who utilized physical activity monitoring (M = 18.1, SD = 17.5 Godin score) technology reported greater average vigorous physical activity scores than those who did not use activity monitoring devices (M = 11.5, SD = 22.3 Godin score). There were no additional significant main (F(1, 402) = 0.11, *p* = 0.74, partial η^2^ < 0.001) or interaction effects (F(1, 402) = 0.08, *p* = 0.78, partial η^2^ < 0.001, Table 1).

### 3.3. Moderate Physical Activity

There were no significant main (F(1, 402) ≤ 0.07, *p* ≥ 0.80, partial η^2^ < 0.001) or interaction (F(1, 402) = 2.8, *p* = 0.10, partial η^2^ = 0.007) effects for differences in moderate physical activity (Table 1).

### 3.4. Mild Physical Activity

For differences in mild physical activity, there was a significant main effect of group (F(1, 402) = 5.4, *p* = 0.02, partial η^2^ = 0.013). This difference was due to greater mild physical activity scores in participants who did not utilize physical activity monitoring technology (M = 12.0, SD = 14.7 Godin score) versus those who did use activity monitoring technology (M = 9.1, SD = 7.9 Godin score). There were no additional significant main or interaction effects (F(1, 402) = 2.1, *p* = 0.15, partial η^2^ = 0.005, Table 1).

### 3.5. Sedentary Behavior

There was a significant main effect of time (F(1, 402) = 76.0, *p* < 0.001, partial η^2^ = 0.16) for differences in sedentary behavior. This difference was due to an increase in sitting from pre-pandemic (M = 2896.7, SD = 1245.6 min/week) to after the cancellation of face-to-face classes (M = 3387.5, SD = 1466.1 min/week). There was also a significant main effect of group (F(1, 402) = 7.0, *p* = 0.009, partial η^2^ = 0.017). This difference was due to greater sitting in participants who did not utilize physical activity monitoring technology (M = 3281.8, SD = 1262.0 min/week) versus those who did use activity monitoring technology (M = 2953.7, SD = 1199.4 min/week). There did not exist a significant group by time interaction (F(1, 402) = 1.6, *p* = 0.21, partial η^2^ = 0.004, Table 1).

### 3.6. Weight

There was a significant main effect of time (F(1, 402) = 12.8, *p* < 0.001, partial η^2^ = 0.03) for differences in bodyweight. Participants reported increased bodyweight from pre-pandemic (M = 79.2, SD = 22.7 kg) to after the cancellation of face-to-face classes (M = 79.9, SD = 23.1 kg). There was also a trend towards a main effect for the group (F(1, 402) = 3.7, *p* = 0.057, partial η^2^ = 0.009) as those who used activity monitors (M = 77.0, SD = 19.0 kg) reported a lower average bodyweight than those who did not use monitors (M = 81.4, SD = 25.2 kg). There did not exist a significant group by time interaction (F(1, 402) = 0.2, *p* = 0.69, partial η^2^ < 0.004, Table 1).

## 4. Discussion

The present study appears to be the first study to assess the ability of the use of physical activity monitoring technology to mitigate potentially negative changes in physical activity, sedentary behavior, and bodyweight during the COVID-19 pandemic in a sample of university students and employees. In the present study, participants who used activity monitors reported 57% greater vigorous physical activity, 11% less sedentary behavior, and 6% lower bodyweight on average than participants who did not use this technology. However, there were no differences between groups for moderate and total physical activity and those that did not use activity monitoring participated in 32% more mild physical activity behavior than those who used monitoring. Additionally, from pre-cancellation to post-cancellation of face-to-face classes, there were no significant changes in physical activity but there was a significant increase in sedentary behavior of 17% and a small but statistically significant increase of 0.59 kg bodyweight. More importantly for the present study, the use of activity monitoring technology did not moderate the changes of any of these dependent variables from pre-cancellation to post-cancellation of face-to-face classes. This suggests that the use of activity monitoring technology did not help prevent negative pandemic-related outcomes (i.e., increases in sedentary behavior and bodyweight).

There is emerging evidence that changes associated with pandemic-related restrictions and recommendations may have resulted in increased sedentary behavior, decreased physical activity, and increased bodyweight [16,17,18,19,20,21,22,23,25]. This is concerning not only because each of these changes are associated with increased risk of cardiometabolic disease but there is also evidence that suggests excess bodyweight may be associated with greater illness severity should an individual contract COVID-19 [13,26,27,28,29,49,50]. The significant increases in sedentary behavior and bodyweight noted in the present study provide additional evidence that restrictions associated with the COVID-19 pandemic may have negatively impacted health behaviors and may have contributed to weight gain. However, while several studies have reported decreases in physical activity during the pandemic, there were no such changes in the present study [16,20,21,23]. It is important to note that the prior evidence of the effects of the pandemic upon physical activity are somewhat equivocal. Research from our group and others has reported that pandemic-related changes in physical activity may be related to pre-pandemic levels of physical activity participation. In these prior studies, individuals who were highly active before the pandemic reported significant decreases in physical activity whereas those who were less active reported increases [20,36]. Despite these differential responses, when the highly active and less active groups were examined in aggregate there was no significant change in physical activity from pre to post implementation of pandemic related restrictions. Presently our two groups (with or without monitors) reported similar total physical activity prior to the pandemic. Therefore, as was the case in prior studies, the lack of change in physical activity across time in the present study may then be due to these pre-pandemic group similarities.

While there is evidence that using activity monitoring may have a positive impact on physical activity and sedentary behavior and that individuals who use such technology may be more active and less sedentary than those who do not use this technology, this evidence is equivocal. Multiple studies have reported that activity monitoring promoted greater physical activity in the user and work from our group demonstrated that these devices may have utility in helping individuals maintain physical activity over time [31,32,33,34]. Conversely, other studies have reported that using activity monitoring provided no benefits as part of an exercise intervention and that self-selected use of activity monitoring technology is only a marker for an elevated exercise identity [35,36]. Because of these equivocal findings, it was not clear if such technology would be beneficial in the context of the current study. However, participants in the present study, regardless of group, significantly increased sedentary behavior and bodyweight from pre-pandemic to post-cancellation of face-to-face classes. Therefore, using physical activity monitoring technology during the COVID-19 pandemic provided no benefit as it pertains to sedentary behavior and bodyweight maintenance.

While this study provides novel information regarding the potential impact of physical activity monitoring technology on health behaviors during the COVID-19 pandemic, it is not without limitations. The sample consisted only of individuals from a single large public university in the American Midwest. This limits our ability to generalize the results of the present study to other populations. The study also consisted of self-report survey instruments which required participants to recall their behaviors and bodyweight prior to the cancellation of face-to-face classes. While this is recognized as a current limitation, there is evidence supporting the validity of the present survey instruments and the use of recall [38,39,40,41,45,46,47]. In fact, there is evidence that individuals can accurately recall health behaviors and bodyweight from far longer in the past (i.e., years) than the participants in the present study were required to (i.e., two–three months) [42,43,44]. Furthermore, the cancellation of face-to-face classes could not have been anticipated and in-person data collection was not possible at the time this survey was administered. Therefore, despite their limitations, the present methods represented a viable approach for assessing these outcomes given the restrictions associated with the pandemic. Another limitation of the current study was that we did not attempt to assess changes in dietary behavior during the pandemic, which may have played a role in the weight gain noted in the present study. Lastly, physical activity technology use was assessed only as a “yes” or “no” proposition. There was no assessment of the amount of activity monitoring technology use within the group that did regularly use this technology.

## 5. Conclusions

This study adds to the growing body of evidence that the initial months of the COVID-19 pandemic may be associated with increased sedentary behavior and bodyweight. More importantly, this study appears to provide the first evidence that the use of physical activity monitoring technology did not mitigate these negative outcomes. While there were some overall group differences in vigorous and mild physical activity, physical activity monitoring technology did not provide any greater protection against the pandemic-related changes with respect to sitting or bodyweight in the participants who utilized this technology versus those that did not. In conclusion, as pandemic-related restrictions persist and/or future restrictions are implemented, it is important for individuals to be mindful of their health behaviors and, especially, to take steps to minimize sedentary behavior. Unfortunately, the simple act of utilizing physical activity monitoring technology does not appear to aid in this regard.

## Figures and Tables

**Table 1 ijerph-18-07518-t001:** Descriptive statistics of dependent variables by group.

Dependent Variable	Group	Pre-Cancellation	Post-Cancellation
Total physical activity (Godin score)	Used monitors (*n* = 172)	41.8 ± 27.3	43.8 ± 29.2
	No monitors (*n* = 232)	41.5 ± 53.4	37.3 ± 43.1
	Total (*n* = 404)	41.6 ± 44.1	40.1 ± 37.9
Vigorous physical activity (Godin score)	Used monitors *	18.1 ± 20.9	18.1 ± 20.0
	No monitors	11.8 ± 23.2	11.3 ± 23.8
	Total	14.5 ± 22.4	14.2 ± 22.5
Moderate physical activity (Godin score)	Used monitors	14.6 ± 13.1	16.6 ± 15.5
	No monitors	16.7 ± 28.8	15.2 ± 20.2
	Total	15.8 ± 23.4	15.8 ± 18.3
Mild physical activity (Godin score)	Used monitors *	9.1 ± 10.7	9.1 ± 7.6
	No monitors	13.1 ± 20.1	10.9 ± 13.2
	Total	11.4 ± 16.9	10.2 ± 11.2
Sedentary behavior (min/week)	Used monitors *	2747.7 ± 1239.2	3159.7 ± 1407.8
	No monitors	3007.1 ± 1241.6	3556.4 ± 1488.5
	Total †	2896.7 ± 1245.6	3387.5 ± 1466.1
Bodyweight (kg)	Used monitors	76.8 ± 18.6	77.3 ± 19.4
	No monitors	81.1 ± 25.2	81.8 ± 25..5
	Total †	79.2 ± 22.7	79.9 ± 23.1

Data are mean ± SD. * significant main effect of group (*p* ≤ 0.02). † significant main effect of time (*p* < 0.001). There were no significant interactions (*p* ≥ 0.10).

## Data Availability

The data presented in this study are available on request from the corresponding author.

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
