# Peer review of "Impact of Activity Monitoring on Physical Activity, Sedentary Behavior, and Body Weight during the COVID-19 Pandemic"

_ijerph, 2021, doi:10.3390/ijerph18147518_

Round 1

Reviewer 1 Report

The present study aims to assess the ability of physical activity monitoring technology to moderate any pandemic-related changes in physical activity, sedentary behavior, and body weight, and to assess potential changes during the COVID-19 pandemic. Although the authors have a good intention and idea, the methodology carried out lacks scientific rigor to pretend to obtain the objective of the study, based on a very subjective and unreliable data collection, for what unfortunately I consider (and of course, I can be wrong) that it is not advisable to publish this type of study.

Specific comments:

Line 11. I recommend not using the abbreviation yrs.

Weight measurements must be reported in Kg, according to international recommendations.

I recommend not using "they", "we". Best passive in the whole manuscript.

Sitting? It is advisable to speak of sedentary behavior.

Line 66-67 "One possible way to prevent decreases in physical activity and increases in sedentary behavior is the use of physical activity monitoring technology." You need add a reference for this sentence.

Line 76-87. I suggest eliminating the hypothesis and leaving only the objective of the work

Line 120-122. The IPAQ questionnaire (reference 39), is designed for "last 7 days" or a "usual week". Therefore, the argument that the authors indicate here to measure past and present physical activity is not valid. Reference 38 does not apply here.

Filling out a physical activity questionnaire requesting information from 2/3 months ago is a problem. Moreover, evaluating physical activity and sedentary behavior through questionnaires is a subjective method. Adding to this subjectivity, the problem of carrying out the questionnaire after a long time represents a serious problem in the truthful interpretation of the data reported by the participants.

Line 139. Here is an error in the date.

Line 144-149. Sedentary behavior does not just involve sitting. Measuring sedentary behavior with this question is not methodologically correct. Also I do not understand how it is written, because it is said that the IPAQ was used, and then he was asked this. The wording in this paragraph should be improved.

Line 150-152. Reference 44 is said to report validation of self-reporting bodyweight in adults, however reference 44 was performed with adolescents aged 13 to 17 years. In addition, it is not validating, taking two measurements of the weight at different times, asking the current one and the one from a few months ago. This, as happened with the measurement of physical activity and sedentary behavior three months ago, is a serious problem, and lacks scientific rigor.

Line 153-157. The fact of regularly using a physical activity monitor does not imply keeping track of physical activity, or having extra motivation to differentiate between two groups. In this sense, using this device does not imply obtaining useful information to assess the ability of physical activity monitoring technology to moderate any pandemic-related changes in physical activity, sedentary behavior (objective of the work). They were asked, if they downloaded or looked at the information of the devices efficiently to control their physical activity, if they wore it all day ... This type of information could have been useful, to validate its use.

Line 183. Mild Physical Activity, must be reported as light physical activity.

Section 3.2. It should be eliminated since the data shown here has been previously written.

Author Response

Thank you for your valuable feedback. We have provided a point-by-point response to each of your comments in the attached document. Changes resulting from your feedback can be seen by reviewing our revised manuscript with “track changes.” Should you have any additional questions and/or feedback, please do not hesitate to let us know.

Sincerely,

J.B. and G.F. 

Reviewer 2 Report

Major comments:

This study investigated the effects of wearable activity trackers on physical activity (PA) levels and body weight during the COVID-19 pandemic. The authors reported participants who use such devices had 57% greater vigorous physical activity, 11% less sedentary behavior, and 6% lower body weight on average than participants who did not use such devices. However, there were no differences between groups for total physical activity levels. Based on these, the authors concluded that the use of wearable activity trackers could not help prevent negative pandemic health impacts. The results from the present study are valuable and of great interest to exercise practitioners and researchers in exercise and health sciences. However, this manuscript has some concerns to be addressed.

Although the results of this study are based on the use or not of wearable trackers, the authors did not show that participants how frequently use such devices in a week. Thus, the reviewer recommends the authors show additional data for adherence rate to such device use, because the reviewer and other researchers want to understand whether adherence rate to such devices moderates preventive effects on negative pandemic health impacts. 

Additionally, the authors should add the reasons for not using IPAQ for PA surveys. And, why the authors don't use both questionnaires (i.e., Godin and IPAQ) with cross-validation for investigating PA levels?

Minor comment:

Please use SI units, “kg,” in the manuscript, instead of “lbs.”

I hope these comments will be helpful.

Author Response

(The authors gave the same response as above.)

Reviewer 3 Report

First of all, I would like to congratulate the authors for the interesting work. This article was a good read. The manuscript is well written and structured, providing a specific scientific background for a proper reader’s comprehension. The topic is emerging and yet “unexplored”, so I find this work can provide some new interesting insights for the scientific community.

The COVID-19 pandemic has seriously challenged our lifestyle and the public health and economy of the entire world. Understanding the effects on different health outcomes of the pandemic-related restrictions (e.g. home confinement, social distancing, educational centers closures, etc.) may help governments to adopt better and more efficient strategies and preventive measures in the future. The present study has two main objectives: 1. To investigate the effects of the pandemic-related restrictions on physical activity levels, sedentary behavior and bodyweight in a university sample in the USA; and 2. To evaluate if the fact of using physical activity monitoring technology can reduce the negative changes in lifestyle associated with the COVID-19 pandemic.

I have only a few questions, suggestions and comments, which I hope can help to improve the quality of this article.

Introduction:

Lines 44-45: It depends on the countries’ restrictions characteristics, like in some European countries (e.g. Italy, Spain, etc.) where the COVID-19 pandemic more fiercely hit. Please, see below the comments referred to lines 237-241.

Line 56: Please remove “that” at the beginning of the sentence.

I suggest to include in the introduction section the concept of “active communities”.

Materials and Methods:

Line 118: For how long were the stay-at-home orders issued? Could people exercise outdoors at that time? Maybe you can add a line to specify this information.

Lines 122-124: Please also provide this information in the “Institutional Review Board Statement” section and in the “Informed Consent Statement” section, at the end of your manuscript.

Line 158: I suggest to change it to “Statistical analyses”.

Results section:

Did you compare data obtained from the different categories of your sample (i.e. faculty, staff, undergraduate, and graduate students)? It would be interesting to know if the pandemic-related restrictions affected in the same way the physical activity patterns, sedentary behavior and bodyweight of the aforementioned groups. It would be also of interest for you to explore if the use of physical activity monitoring technology was similar in these groups. Maybe younger people (e.g. undergraduate and graduate students) are more familiarized with such technology/devices?

Table 1: Please provide a title for Table 1. I also suggest authors “to combine rows” for each dependent variable for providing more clarity and creating a better visual impact. In this sense, I think that reader’s comprehension would be easier if it is clear that, for example, “Total physical activity” category is inside one cell and contains the other three cells (i.e. used monitors, no monitors, total) in the adjacent column.

Discussion section:

Lines 237-241: In my opinion, the fact that pandemic-related restrictions can affect physical activity levels may be influenced by the kind of restrictions itself and by the countries’ management of these restrictions. At the time of your study, in some European countries like Spain or Italy governments imposed the lockdown of the countries and people could only leave their homes to get medical care, go to the pharmacy or go grocery shopping. This means that the possibility to exercise was limited and home-based. On the other hand, other countries, like France, allowed citizens to exercise outdoor for one hour, once a day and within a distance of one km. If we compare both situations, it is reasonable to think that changes in physical activity levels and lifestyle could be linked to the kind of restrictions imposed by each country. In this sense, regions with active communities could also register higher physical activity levels as it is “easier” for people to be physically active.

Maybe you want to add a couple of lines with these ideas in your discussion section…

Lines 267-279: I think another limitation of your study, that could have played a role in bodyweight gains, was a lack of information of the sample nutrition patterns. Did these patterns change across the pandemic-related restrictions period?

References:

Please revise the IJERPH instructions for authors.

Like I said above, these few comments only aim to improve the quality of your work.

Author Response

(The authors gave the same response as above.)

Round 2

Reviewer 1 Report

I appreciate that you have provided references to justify the use of using the questionnaires after a while, regardless of the fact that they are references from three decades ago and not very relevant with the passage of time, I add a series of arguments which justify my position. I also appreciate the corrections you have made according to my comments. I continue to think that the limitation is very serious, not recommending the publication of this article.

I disagree that the methodology used to classify the use of the activity devices is correct. This work would have improved in quality, if physical activity had been objectively measured at the right time. I suggest that the editor considers the invitation of additional reviewers for this manuscript.

It is well known that the COVID situation was unpredictable, but we cannot justify the use of non-validated tools for a past time due to the fact of an unforeseeable situation.

The correlations of the works that it contributes are really low, something about this appears in the following information that I contribute.

I recommend you read this much-cited review work (http://dx.doi.org/10.1136/bjsm.37.3.197), belonging to the journal with the greatest impact in the area of ​​sport sciences.

From which I extract some aspects:

“Because of  limitations in human memory, the reliability of information generally decreases with the length of the period surveyed, and it is best to keep the reporting interval relatively short (no longer than three months4);

“Most authors have looked at indices of total activity. Reliability diminishes with the length of the recall period. Again, this has been assessed by test/retest correlations. Lamb and Brodie102 found a two week coefficient of 0.86, and the five week coefficient for the Minnesota leisure time physical activ- ity questionnaire was 0.88.123 Studies on the college alumnus questionnaire found r values of 0.72 at one month, falling to 0.3–0.4 over 8–12 months.14 32 124 125 Other authors have reported coefficients of a similar order: 0.58–0.67 for the com- munity health activities model program for seniors (CHAMPS) physical activity questionnaire over a six month interval,126 0.55 for adolescents over one year,127 and 0.59 for a two to three year recall in the coronary artery risk development in young adults (CARDIA) study.1 “

“Lack of reliability is due in part to seasonal and/or temporal variations in physical activity patterns, but shortcomings of human memory are also an important problem. Thus, questionnaire responses show a variation of 50% or more, even if one year activity patterns are reassessed after an interval of a few days. The problem is particularly acute if intensities of effort are low.129”

It was suggested to delete reference 38, but it still appears in the text. It has no relevance to the referenced information.